# CatVLM: Enhancing Temporal Understanding in Cataract Surgery Videos with Boundary-Aware VLM

**Jay N. Paranjape**[1] (ID)                                               JPARANJ1@JHU.EDU

**Nisarg Shah**[1]                                                   SNISARG812@GMAIL.COM

**Nanthini Narayanan**[1]                                    NANTHINII.NARAYANAN@GMAIL.COM

**Shameema Sikder**[2]                                                  SSIKDER1@JHMI.EDU

**S. Swaroop Vedula**[1]                                                 SWAROOP@JHU.EDU

**Vishal M. Patel**[1]                                                    VPATEL36@JHU.EDU

[1] *Johns Hopkins University*
[2] *Wilmer Eye Institute*

**Editors:** Under Review for MIDL 2026

## Abstract

Recent studies have shown the effectiveness of Vision Language Models (VLMs) for understanding and analyzing videos in the medical domain and supporting various Question-Answer (QA) tasks. Yet, current VLMs fall short in addressing queries that require temporal reasoning—a critical capability for surgical video understanding. In this work, we introduce CatVLM, a boundary-aware VLM, designed to capture temporal dynamics in untrimmed cataract surgery videos. CatVLM is capable of performing three clinically relevant tasks that demand moment-level awareness: Video Moment Retrieval (VMR), Video Captioning (VC), and Counting. To facilitate the training of such a model, we generate a bank of QA annotations for each task and propose a method to integrate video clips with the timestamps they occur. To the best of our knowledge, this work is one of the first approaches to explicitly incorporate temporal boundary awareness into VLMs for cataracts as well as the medical domain. We evaluate CatVLM on two public cataract surgery datasets, establishing new baselines across all three tasks. All the code, model checkpoints and annotations will be released post-review

**Keywords:** Cataract, Vision Language Models, Video Understanding.

## 1. Introduction

Cataract surgery is one of the most frequently performed operative procedures worldwide (Kauh et al., 2016; Rossi et al., 2021). In recent literature, various deep learning based approaches have been used for intra-operative and post-operative analysis of cataract surgery videos through various downstream tasks. These include skill assessment (Hira et al., 2022; Kim et al., 2019), step and instrument localization (Shah et al., 2023a; Paranjape et al., 2023), segmentation (Ghamsarian et al., 2024) and more recently, comprehensive understanding using vision language models (VLMs) (Ramjee et al., 2024). VLMs can process natural language queries and integrate them with video content to generate clinically relevant answers. However, current approaches are typically trained to produce summarized

Figure 1: We develop CatVLM, a boundary-aware LLM for cataract surgery that is capable of performing temporal video understanding tasks like Video Moment Retrieval (VMR), Video Captioning (VC), and Counting.

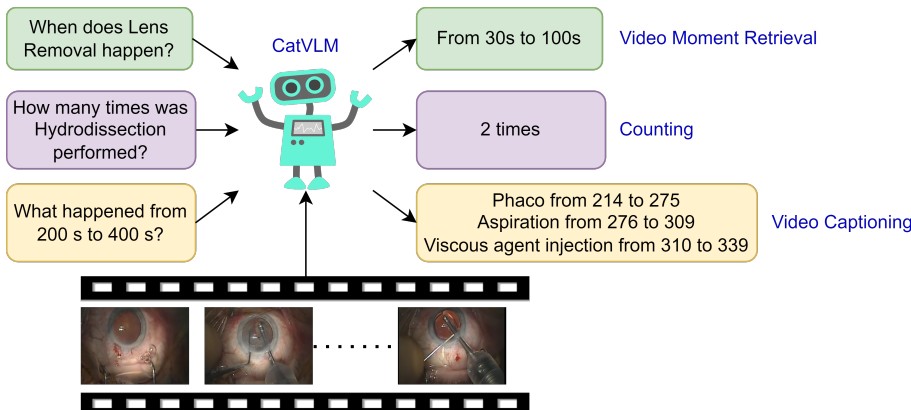

responses, lacking the fine-grained temporal reasoning needed to identify when key surgical events occur, which is essential for medical video analysis. In this work, we introduce CatVLM: a VLM that has improved temporal awareness for analysis of cataract surgery videos. Temporal understanding of surgery videos enables multiple downstream clinical applications. Examples of potential applications include retrieval of relevant segments of the video for analytics such as operative time and surgical skill, characterizing surgery based on how frequently a step is performed or sequence of activities performed during the procedure, and using the analytics to provide feedback to surgeons and enable context-aware systems. Moreover, temporally grounded QA can accelerate expert annotation by allowing annotators to focus directly on relevant video segments rather than reviewing entire procedures. To facilitate this in the cataract surgery domain, we first use the Video-Masked Auto Encoder (VMAE) strategy ((Tong et al., 2022b)) to pretrain a feature extractor on a corpus of cataract surgery videos collected from our hospital. Next, we leverage step localization captions on public datasets to generate a corpus of QA annotations spanning three temporal tasks: VMR, VC and Counting. We then use the frozen encoder to extract video features from the public videos and generate embeddings, which capture the spatio-temporal features of the video, but are still unaware of what explicit timestamp each feature represents. To incorporate this information, we propose the simple yet effective Image-Timestamp (IM-TS) Adapter module, which concatenates timestamp embeddings with video features and refines them through a learnable linear layer to produce time-aware representations. These enriched embeddings are passed to an open-source VLM, which we fine-tune using Low-Rank Adaptation (LoRA) ((Hu et al., 2022)). To further improve task specialization, we employ separate LoRA modules for each of the three tasks. In addition, we learn a lightweight K-Nearest Neighbours (KNN) module that decides which LoRA to use based on the question type. During inference, the user can manually select the LoRA module or utilize the KNN for automatically selecting the LoRA module to plug into the VLM. Thus, our contributions are threefold:

1. We present CatVLM, a boundary-aware VLM capable of answering temporally grounded questions in cataract surgery videos as shown in Figure 1, enabled by VMAE pretraining and the proposed IM-TS Adapter.
2. We develop a method to generate annotations to train such an VLM from public cataract video datasets.
3. We show that our method is able to outperform existing methods for these tasks. Thus, we establish a new baseline for future research.

## 2. Related Work

**Boundary-Aware VLMs in Literature:** In the natural vision domain, several recent works have explored empowering VLMs with the ability to capture video moments (Huang et al., 2024; Maaz et al., 2024; Ren et al., 2023). VideoChatGPT (Maaz et al., 2024) uses frozen CLIP features and trainable adapter layers to extract features, before passing it to the frozen Vicuna (Chiang et al., 2023) backbone. TimeChat (Ren et al., 2023) introduces a time-aware Q-former architecture, combining multiple video Q-formers, adapters, and LoRA modules to make the VLM sensitive to temporal boundaries. While effective, this approach requires training a large number of modules, demanding significantly more data than is available in the cataract surgery setting. On the other hand, VTimeLLM (Huang et al., 2024) uses a three-stage training approach. In the first stage, an adapter is pretrained with natural images. Next, it uses LoRA to tune the model to identify and localize all events in a given video. Finally, in the third stage, an additional LoRA is used to teach the model to respond to natural text using an instruction tuning dataset. In our work, we initialize the LLM with LoRA weights from VTimeLLM, which eliminates the need for instruction tuning and enables open-set question answering rather than template-based QA. Despite these advances, existing approaches face two critical challenges: (1) The feature extractor of CLIP is trained on natural images and is not the best choice for extracting features from cataract videos, and (2) Various important surgical moments in cataracts often last for only a few seconds before transitioning to the next evolution. In the current state, these VLMs aren't able to capture such quick changes and distinguish them as separate key actions.

To address these limitations, we pretrain a Video-MAE (VMAE) (Tong et al., 2022b) feature extractor on a large corpus of private cataract videos from our hospital and use the pretrained model to extract features during training, thus initializing the model with a representative set of features. In addition, unlike existing methods which predict timestamps at a scale of 100, we design our generated annotations such that the start and end times of moments are represented on a scale of 1000, essentially encouraging the model to predict event boundaries at ten times the resolution of existing methods. This allows it to capture steps that are changing quickly.

**Video QA in the Medical Domain:** Recent advances in medical video understanding have focused on integrating VLMs for answering clinically relevant questions including surgical tool detection, procedural understanding and step recognition. This has led to datasets like MedVidQA (Gupta et al., 2023), that contains QA pairs for generic medical videos, and SurgPubVideo (Li et al., 2025), that evaluates conceptual understanding in surgical videos. This has, in turn, enabled various systems performing video QA in the medical domain. (Tascon-Morales et al., 2023) introduced transformer-driven architectures for sur-

gical video QA while Surgical-VQA (Yuan et al., 2024) used hierarchical graph networks and MedicalGPT (Xu, 2023) explored integration of VLMs for allowing open-set question-answering. However, existing datasets and methods primarily focus on multiple-choice or spatial/conceptual questions, overlooking tasks that require temporal localization such as video moment retrieval or captioning. To the best of our knowledge, no datasets currently address temporal video QA specifically in the cataract surgery domain, underscoring the need for temporally aware approaches tailored to this setting.

Figure 2: Model Architecture. The pretraining phase involves self-supervised pretraining on a large corpus of private videos and is responsible for creating a robust feature extractor. The finetuning phase improves these features by integrating timestamp-aware embeddings and passes them to a VLM, which is trained using LoRA for each task. During inference phase, the relevant LoRA module is selected from a bank of LoRA modules trained in the finetuning phase based on the task and used to get the final prediction.

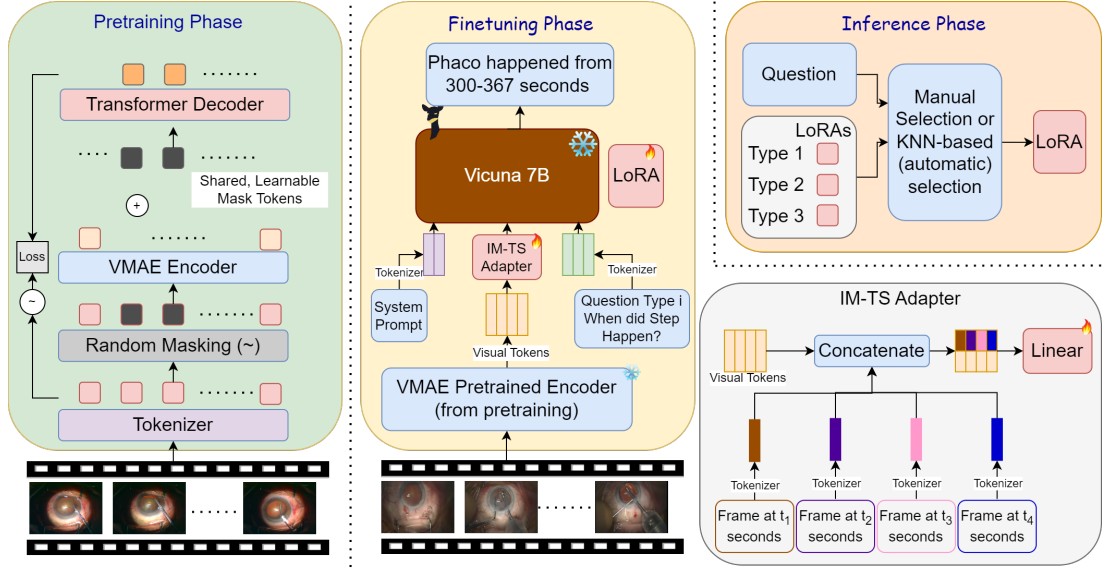

## 3. Methodology

### 3.1. Generating Annotations

We leverage step-localization annotations of public datasets to generate question-answer pairs for tuning CatVLM. Given a video, step localizations cover what steps are present in the video and their time boundaries. Based on these annotations, we generate GPT-generated QA pairs to train CatVLM. In this work, we focus on three important question types as follows:

1. **Video Moment Retrieval:** Q: When did $s$ happen? A: From $t_1$ to $t_2$, from $t_3$ to $t_4$
2. **Video Captioning:** Q: What happened between $t_1$ and $t_2$? A: $s_1$ from $t_1$ to $t_2$, $s_2$ from $t_3$ to $t_4$, ...

3. **Counting:** Q: How many times did $s$ occur? A: $s$ occurred $n$ times

Here, $s$ represents the step name, $t_i$ represents time instances and $n$ represents the number of times a given step was performed at separate intervals. To ensure consistency across videos of varying lengths, all timestamps are normalized to a scale of 0–1000, and are calculated as follows: $t_i = round(1000 \times T_i/D)$, where $T_i$ is the timestamp in seconds and $D$ is the total duration of the videos in seconds. This allows the model to cater to untrimmed videos of any duration since all timestamps are scaled to the same range. For each of these question types, we used ChatGPT to generate 50 synonymous sentence variations. Hence, for a given video, we generate $50 \times total\_steps$ annotations, significantly expanding the training corpus. VMR allows the model to learn the correlations between the visual and temporal boundaries of the given step. Similarly, Video Captioning is a more complex task since it requires the model to identify all the steps happening between two timestamps. Counting is a derived task that allows the model to learn to distinguish between the same events at different points in time. All these tasks aim to enhance the temporal understanding of the LLM and are essential for an in-depth analysis of the video.

### 3.2. Pretraining Phase

The pretraining phase is based on the VideoMAE(Tong et al., 2022a) framework as shown in Fig. 2, and consists of a Tokenizer, Encoder, and Decoder. Given an input video $X \in \mathbb{R}^{T \times C \times H \times W}$, the Tokenizer utilizes a 3D convolution layer followed by positional encoding to convert $X$ into tokens. A fraction of these tokens is then randomly selected based on a predetermined masking ratio. The selected tokens are passed through a Vision Transformer-based encoder to extract latent representations that capture high-level semantics of the input video. The decoder then merges these latent features with fixed representations of the masked tokens - augmented with positional encodings - and uses a transformer architecture to reconstruct the missing tokens. The model is optimized by minimizing the reconstruction error, measured as the Mean Squared Error between the reconstructed video tokens and the original normalized video tokens.

### 3.3. Finetuning Phase:

Once the feature extractor is trained, we freeze its weights and utilize it to extract features from untrimmed videos, as shown in Figure 2. These features capture the spatio-temporal structure of the video but lack explicit alignment with text-based timestamps. Moreover, since videos vary in length, the same set of features can correspond to different temporal positions. To address this, we propose a simple yet effective solution called the IM-TS Adapter. Given the duration $D$ of the video $v$ and its VMAE features $f_v \in \mathbb{R}^{N \times H}$, where $N$ are the number of features and $H$ is the hidden dimension of the features, we compute the timestamp embeddings for each feature as follows:

$$f_t^i = VLM\left(\frac{i \times D}{N} \text{ ``seconds''}\right) \tag{1}$$

These embeddings are concatenated with the original features and passed through a learnable linear transformation to form the final set of features $f$ as follows: $f = Linear(f_v \cdot f_t)$ Here,

VLM denotes the embeddings generated by the language model for the string representation of the timestamp and $\cdot$ represents concatenation. By concatenating timestamp embeddings with video features, the IM-TS Adapter produces time-aware representations that preserve both visual and temporal context. To adapt the VLM to the enriched features $f$ from the IM-TS module, we utilize Low Rank Adaptation (LoRA) (Hu et al., 2022) to perform parameter-efficient fine-tuning. This method allows us to adapt the LLM without tuning all parameters of the large model. We use a separate LoRA for each of the three tasks to allow for more stability during training. The role of the IM-TS adapter is to capture and integrate the timestamp information in each feature, while the LoRA module adapts the LLM to use this information to answer a particular question type. To encourage this role separation, we use the same IM-TS adapter across the three tasks with weight sharing, whereas each LoRA is task-specific.

### 3.4. Inference Phase:

We learn different LoRA modules for different types of tasks, namely VMR, Counting and VC. During inference, we choose which LoRA module to use based on the question, attach it to the VLM and produce the prediction. The choice of LoRA can be manual or automatic depending on the application. In the manual approach, the user provides the LoRA module to use as an input. In the automatic approach, we use a K-Nearest Neighbour (KNN) to classify the question into one of three types and select the LoRA corresponding to the prediction of the KNN. To train the KNN, we generate 50 new synonymous sentences for each question and convert them into embeddings using CLIP (Radford et al., 2021). With this method, we achieve a 97% accuracy in LoRA selection, which is also reflected in our results. An alternative design is to use a single LoRA module for all tasks. However, this approach presents two major limitations:

1. We observed that the model performed worse on all the tasks as shown in Table 2. This happens because the model often confuses the output of one task for the other or gets stuck in local minimas, leading to unstable training.
2. Adding a single LoRA for all three question types limits the number of tasks that can be performed. In this study, we consider three major video understanding tasks. However, for a new type of application, the entire LoRA would have to be retrained with all training data again, to incorporate the new task.

By contrast, maintaining multiple task-specific LoRA modules divides the training data appropriately and isolates learning patterns. In addition, it makes the system modular, allowing new tasks to be incorporated by adding corresponding LoRA experts without re-training existing modules, ensuring scalability and stability.

## 4. Experiments

### 4.1. Datasets

We use two public cataract surgery video datasets for evaluating our method. Cataract-1K (Ghamsarian et al., 2024) consists of 1000 videos with an average video duration of 7.12 minutes. Since this dataset provides step annotations for only a subset of videos, we

manually annotated the rest of the videos for training our model. We will be releasing these annotations post-review. Annotations were done by study personnel trained with a standard protocol designed by an expert surgeon. Reliability of annotations was assured by each annotator labeling the same set of videos and verification of consistent understanding of instructions, with a tolerance of 1 second for step boundaries. Cataract-101 (Schoeffmann et al., 2018) consists of 101 cataract surgery videos performed by four different surgeons of varying expertise and average video length of 8.67 minutes. We annotated the videos for steps using a similar protocol as that used for Cataract-1k. For pretraining, we use private cataract video dataset from the Wilmer Eye Institute, with 450 videos (Shah et al., 2025). These videos have an average duration of 34 minutes and were originally recorded at 59 *fps*. Following prior studies (Gao et al., 2021; Shah et al., 2023b), all videos are subsampled to 1 *fps* and resized to $250 \times 250$ pixels for pretraining. We ensure that videos used for pretraining are not used during finetuning or inference.

### 4.2. Implementation Details

We used the Vicuna (Chiang et al., 2023) model initialized with the weights from VTimeLLM (Huang et al., 2024) since they train the LLM using instruction-tuning that allows the LLM to recognize instructions from natural language without the requirement of any template. We use Vicuna as the backbone VLM to ensure fair comparison with existing methods for temporal video QA. For the individual LoRA experts, we use a rank of 64 and an initial learning rate of 1e-4. The LoRA was trained for 1 epoch with batch size 8 on a single NVIDIA RTX A6000 GPU. For pretraining, we use AdamW optimizer with a base learning rate of $1 \times 10^{-3}$ and a weight decay of 0.05, using a cosine decay schedule with a warmup period of 40 epochs. Training is performed over 800 epochs on 8 NVIDIA RTX A6000 GPUs with a batch size of 64, where each input consists of 16 frames.

### 4.3. Metrics

We compute the mean Average Precision (mAP), mean Average Recall (mAR) and mean Intersection-over-Union (mIoU) metrics for VMR and VC tasks and mean accuracy for the counting task. For mAP and mAR, we compute the precision and recall at IoU thresholds of 0.3, 0.5 and 0.7 and compute the average of the three values. To calculate the recall and precision, we calculate the total number of predicted timestamps that had IoU greater than the selected threshold with a ground truth and consider them as true positives. The ground truths that were not matched with any prediction are considered as false negatives, while the predictions that were not true positives were considered as false positives.

### 4.4. Results

We compare our method against state-of-the-art (SOTA) methods from the natural vision domain, which we finetune on the cataracts training data. As seen in Table 1, we outperform the current SOTA methods for temporal LLMs on all three tasks for Cataracts-101 and for VC and VMR on Cataract-1K with comparable performance on the Counting task. The task of Counting is simpler than VC or VMR and so we see greater than 85% accuracy in all cases, with our method reaching 96% and 89% accuracies on Cataract-1K and Cataracts-101 respectively. With respect to VTimeLLM, for the task of VMR, we see a rise in 6% for

Cataracts-101 and 4% for Cataract-1K. For VC, we see a rise of 4% in Cataracts-101 but do not see significant change for Cataract-1K. We see a slight drop in metrics going from the manual LoRA selection to the automatic LoRA selection case, which is expected since errors from the KNN are also reflected in the results for the automatic case. It is important to note, however, that manually selecting the LoRA is not infeasible during inference.

In addition, we note that there is room for improving these numbers further. We observed that this is caused due to limited size of the dataset, causing overfitting during tuning the model. This also points towards the challenging nature of these two tasks, especially in the case of limited annotations and motivates further research in this field. Figure 3 shows qualitative results of our method for VMR (a) and VC (b). As seen in the figure, our method performs precise predictions for these tasks, and is able to identify boundaries correctly for all steps. Part c in the figure shows a failure case of our model, where one event boundary is correctly predicted but it misses the other occurrence by some seconds.

Table 1: Quantitative Results on two public datasets. The first question represents VMR, the second represents Counting and the third question denotes VC. All results have a maximum p-value of 1e-5 with our method, making it statistically significant.

| | Q: When did $s$ happen? | | | Q: How many times did $s$ occur? | Q: What happened between $t_1$ and $t_2$? | | |
|---|---|---|---|---|---|---|---|
| Method | mAR | mAP | mIoU | Accuracy | mAR | mAP | mIoU |
| **Cataracts-101** (Schoeffmann et al., 2018) | | | | | | | |
| VideoChatGPT (Maaz et al., 2024) | 0.38 | 0.36 | 0.37 | 85.4 | 0.23 | 0.24 | 0.23 |
| TimeChat (Ren et al., 2023) | 0.43 | 0.45 | **0.45** | 86.7 | 0.33 | 0.19 | 0.17 |
| VTimeLLM (Huang et al., 2024) | 0.36 | 0.37 | 0.37 | 85.6 | 0.55 | 0.43 | 0.40 |
| Ours (Manual LoRA selection) | **0.45** | **0.46** | 0.43 | **87.8** | **0.59** | **0.45** | **0.42** |
| Ours (Automatic LoRA selection) | 0.44 | 0.45 | 0.42 | 85.2 | 0.58 | 0.44 | 0.41 |
| **Cataract-1K** (Ghamsarian et al., 2024) | | | | | | | |
| VideoChatGPT (Maaz et al., 2024) | 0.15 | 0.16 | 0.16 | 96.1 | **0.23** | **0.24** | 0.23 |
| TimeChat (Ren et al., 2023) | 0.14 | 0.14 | 0.17 | 95.4 | 0.01 | 0.02 | 0.01 |
| VTimeLLM (Huang et al., 2024) | 0.18 | 0.18 | 0.18 | **96.8** | 0.21 | 0.22 | 0.22 |
| Ours (Manual LoRA Selection) | **0.22** | **0.22** | **0.21** | 95.8 | **0.23** | **0.24** | **0.24** |
| Ours (Automatic LoRA selection) | 0.21 | 0.21 | 0.20 | 92.5 | 0.22 | 0.23 | 0.23 |

Table 2: Ablations for MoE and increasing resolution on Cataracts-101

| Method | range of timestamp | LoRA style | Q: When did $s$ happen? | | | Q: How many times did $s$ occur? | Q: What happened between $t_1$ and $t_2$? | | |
|---|---|---|---|---|---|---|---|---|---|
| | | | mAR | mAP | mIoU | Accuracy | mAR | mAP | mIoU |
| Ablation 1 | 100 | Single | 0.34 | 0.36 | 0.35 | 84.0 | 0.54 | 0.29 | 0.36 |
| Ablation 2 | 100 | Multiple | 0.37 | 0.38 | 0.38 | 86.6 | 0.51 | 0.39 | 0.35 |
| Ablation 3 | 1000 | Single | 0.36 | 0.39 | 0.38 | 85.1 | 0.53 | 0.43 | 0.38 |
| Ours | 1000 | Multiple | **0.45** | **0.46** | **0.42** | **87.8** | **0.59** | **0.45** | **0.42** |

## 4.5. Ablations

**Ablation on Multiple LoRAs and Increased Resolution:** We analyze the effects of our design choices in Table 2 for Cataracts-101 dataset. The first row represents using the

Table 3: Ablations for model components on Cataracts-101

| Method | VMAE features | IM-TS Adapter | Q: When did s happen? | | | Q: How many times did s occur? | Q: What happened between $t_1$ and $t_2$? | | |
|---|---|---|---|---|---|---|---|---|---|
| | | | mAR | mAP | mIoU | Accuracy | mAR | mAP | mIoU |
| Ablation 1 | | | 0.36 | 0.37 | 0.36 | 84.8 | 0.54 | 0.40 | 0.38 |
| Ablation 2 | ✓ | | 0.43 | 0.43 | 0.41 | 86.9 | 0.59 | 0.44 | 0.42 |
| Ablation 3 | | ✓ | 0.37 | 0.38 | 0.38 | 87.0 | 0.59 | 0.44 | 0.41 |
| Ours | ✓ | ✓ | **0.45** | **0.46** | **0.43** | **87.8** | **0.59** | **0.45** | **0.42** |

default range of 0 - 100 for representing the timestamps, which is given by $t_i = round(100 \times T_i/D)$, where $t_i$, $T_i$, $D$ represent the new timestamp, original timestamp and duration of the video respectively. In addition, it uses a single LoRA for tuning. The second row represents the range of 100 but incorporates a multiple LoRA approach, with a separate LoRA for each task. The third row has an increased resolution of 1000, but incorporates a single LoRA. Finally, the last row is the proposed method, which has a range of 1000 and uses multiple LoRA modules. We use the manual LoRA selection process in the case of multiple LoRA modules. As seen in the table, each of the design choices has a non-trivial contribution in raising the performance of the model.

**Ablation on Model Components:** VMAE features and the IM-TS modules are the two major components in our method. To gauge the effectiveness of each of these, we perform an ablation by adding these one by one to the Vicuna model and evaluate performance on Cataracts-101 dataset, as seen in Table 3. In the first row, we use CLIP embeddings, similar to VTimeLLM. Next, we replace the CLIP features with VMAE features in the second row from the pretraining phase of CatVLM, which significantly increases the performance, showing the effectiveness of pretraining. In the third row, we retain the CLIP features and introduce the IM-TS adapter. Similar to row 2, we also see a rise in metrics, showing the effectiveness of the individual modules. Finally, using both components gives the best metrics as seen in the fourth row. We use a timestamp range as 0 to 1000 and multiple LoRAs with manual selection for all the rows.

## 5. Conclusion:

In this work, we present CatVLM, a VLM that is more aware of the finer temporal boundaries of events and is able to perform downstream temporal tasks including Moment Retrieval, Counting and Captioning on untrimmed videos of cataract surgery. CatVLM uses discriminative VMAE features and a novel IM-TS adapter for learning spatio-temporal and timestamp-related features, as well as multiple LoRA experts allowing for a more stable training and the possibility to add more tasks easily without retraining for other tasks. Through our work, we aim to motivate further research in this field by establishing a baseline for three tasks on two datasets. In the current version, CatVLM shows room for improvement for mainly the Video Captioning and Moment Retrieval tasks, mainly due to overfitting. Future directions of research involve extending boundary-aware LLMs to other modalities and adding more tasks to the system.

Figure 3: Qualitative Results. (a) shows the VMR task. Our method captures the step with a high IoU with the GT. (b) represents VC, where the prediction from our model captures all steps with a high IoU. (c) represents a failure case for the VMR task, where one occurrence is captured but the other occurrence is not.

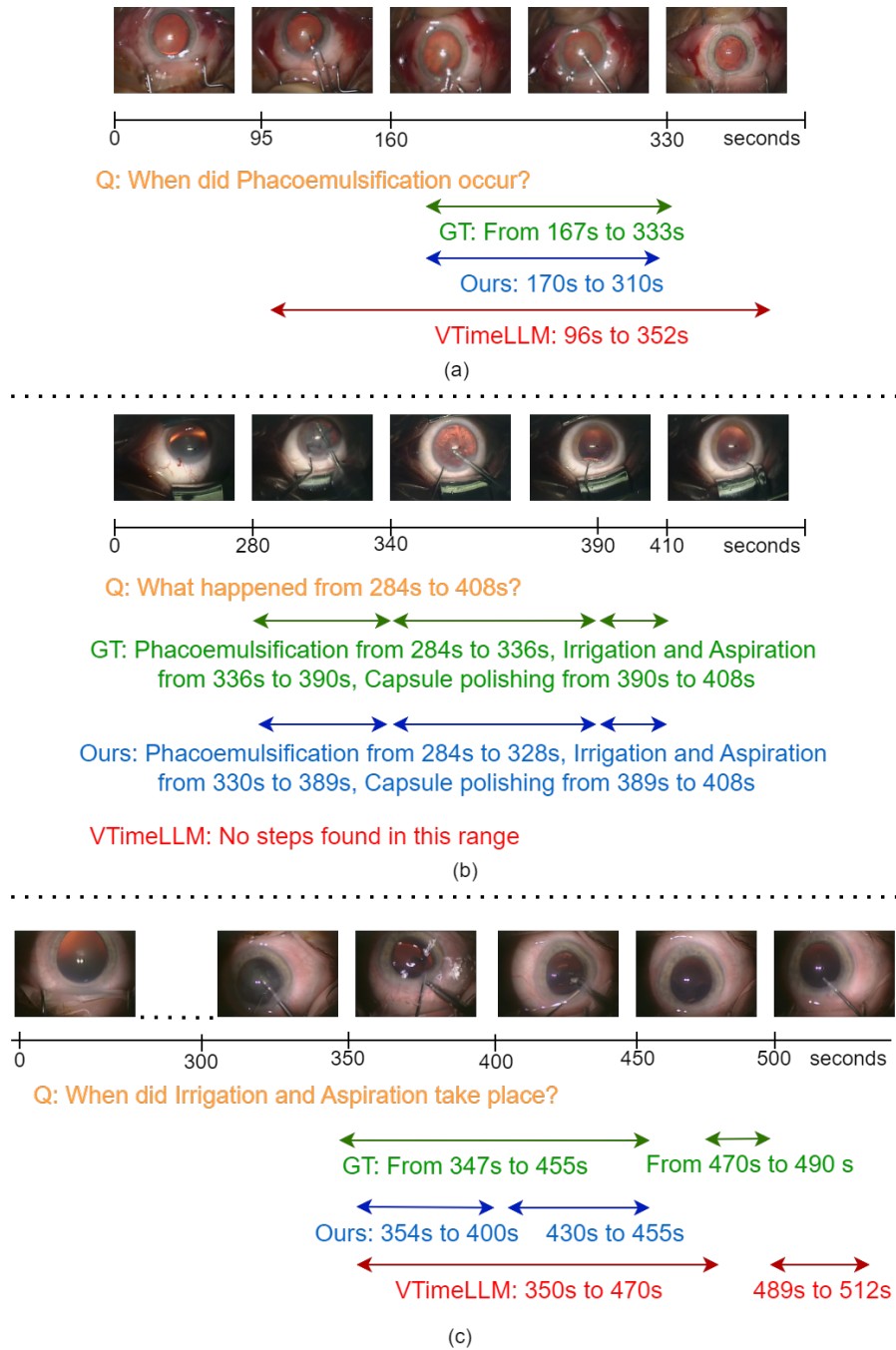

## Acknowledgments

The authors are supported by grants from the National Institutes of Health, U.S.A.; NIH 1R01EY033065 and NIH 1R01EB038734. The content is solely the responsibility of the authors and does not necessarily represent the official views of the National Institutes of Health.

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
