# OpenReview forum: "CatVLM: Enhancing Temporal Understanding in Cataract Surgery Videos with Boundary-Aware VLM"
_MIDL.io/2026/Conference — MIDL 2026 Poster_

### Official Review · Reviewer_yg9x · 2026-01-05

**Confidence:** 3
**Preliminary Rating:** 3
**Final Rating:** 4

**Summary:**

This paper proposes CatVLM, a boundary-aware video-language framework for cataract surgery videos that aims to improve temporal understanding under long-horizon, fine-grained procedural dynamics. The key idea is an IM-TS adapter that injects explicit timestamps into visual segment representations and a multi-LoRA routing strategy that selects task-specific low-rank adapters for heterogeneous tasks (VMR, interval-based narration/captioning” and counting) using KNN to reduce interference. Experiments on Cataract-101 and Cataract-1K report consistent gains over recent Video-LLM baselines, with ablations suggesting benefits from timestamp injection and multi-adapter specialization. Overall, the work is significant for clinical workflow analysis because it targets temporally precise reasoning rather than static frame semantics.

**Strengths:**

- High practical relevance. The paper targets long-horizon temporal understanding in cataract surgery videos (boundaries, ordering, interval queries, counting), which aligns well with real clinical workflow analysis needs.
- Lightweight and intuitive design. The timestamp injection (IM-TS) and task-specific multi-LoRA adapters are conceptually simple, easy to integrate, and also lightweight to be used in practical applications.
- Clear overall presentation. The paper is generally easy to follow, with a coherent structure from motivation to method to experiments and a relevant related-work discussion.

**Weaknesses:**

- Evaluation definitions appear inconsistent/incorrect. The metric section states that unmatched ground truths are treated as true negatives, which is not consistent with standard detection/localization evaluation (they should be false negatives). The paper also does not clearly specify the one-to-one matching protocol (e.g., greedy/Hungarian, IoU thresholds) used to compute.
- Task formulation is not sufficiently clarified, especially for Video Captioning (VC). The VC task seems closer to interval-based step narration with temporal boundaries, yet it is evaluated with detection-style metrics (mAP/mIoU). I am a little confused about the output text-format, and the authors need to justify why the chosen metrics are appropriate for what is called captioning.
- Protocol and comparability details are incomplete. Key dataset and annotation details for Cataract-1K (split policy, annotation process, boundary tolerance, inter-rater agreement) are not fully described.

**Detailed Comments:**

- The VC task definition should be clarified: what exactly is the output format (free-form text vs. step labels with temporal segments), and why detection-style metrics are appropriate for this task. Or I might be missing something. Please clarify it.
- Please specify the exact matching protocol used for IoU-based evaluation (one-to-one matching, IoU thresholds, greedy vs. Hungarian, handling of duplicate predictions).
- For Cataract-1K, add key protocol details (split policy, annotation procedure, boundary tolerance, and any inter-annotator agreement).
- Minor: improve notation consistency in the IM-TS description (explicitly indicate concatenation vs. multiplication) and fix small grammar/consistency issues (dataset naming, capitalization).

**Justification Of Final Rating:**

Based on the clarifications and a few minor corrections from authors, I think the authors have addressed my main questions and concerns on evaluation definition, task formulation, and evaluation details. Therefore, I would raise my evaluation to weak accept.

**Justification Of The Preliminary Rating:**

I lean toward Borderline mainly due to concerns about the evaluation protocol and clarity of task formulation. The metric definition appears inconsistent with standard localization evaluation (e.g., unmatched ground truths described as true negatives) and the matching protocol for IoU-based metrics is not clearly specified, which directly affects the validity and reproducibility of the reported results. In addition, the VC task is not sufficiently defined relative to its detection-style metrics, making it difficult to interpret what is being evaluated. The method ideas are reasonable and the application is relevant, but the current presentation leaves critical ambiguity in how results are computed and attributed. I'd like to raise a higher score if the above issues can be addressed.

**Questions To Address In The Rebuttal:**

- Evaluation correctness: The paper states unmatched ground truths are treated as true negatives. Is this a writing mistake or the actual implementation?
- VC formulation: Is VC free-form captioning or structured step segmentation/narration? Please show a few qualitative output examples and explain how the reported mAP/mIoU is computed for this output.
- Comparability/attribution: How much of the gain comes from timestamp injection vs. private video-feature pretraining vs. multi-LoRA? A concise controlled ablation (same backbone/init/features across comparisons) would help.

---

> ### Author Response · Authors · 2026-01-22
> **1) Formulation of VC task and Metric Calculation**
>
> We thank the reviewer for their detailed comments and try to address them pointwise. We request the reviewer to kindly raise their score if we were able to address their concerns satisfactorily.
>
> As mentioned in Section 3.1, the task of VC takes in a free-form user question that contains two timestamps and outputs the steps happening in between those timestamps. The question can be free-form, for example: "What happened between t1 and t2" and "What steps were seen during the time from t1 to t2". The output is not architecturally constrained to follow a particular format. However, for ease of metric calculation, we use the same format, namely "s1 from t1 to t2, s2 from t3 to t4" and so on, in the training data annotations. This encourages the model to follow the format for the output and we parse it to collect the step names s1,s2 and corresponding time ranges (t1,t2), (t3,t4), respectively.
>
> For evaluating the quality of VC, in general video captioning papers like TimeChat and VTimeLLM, there are two types of metrics. One measures the captioning quality by comparing n-grams with the ground truth and is computed by METEOR score or CIDER score. The other metric measures whether the predicted time ranges match the ground truth ranges, which is computed using mAP and mAR. In our case, we output the step name directly. Hence, there is no requirement for n-gram matching. For a given step in the prediction, if it does not exist in the GT, it is a false positive (FP). Conversely, if a step in the ground truth does not exist in the prediction, its a false negative (FN). For both these cases, the IoU is 0. For steps that exist in both the GT and the prediction, we compute the maximum IoU between their corresponding time ranges using the Hungarian algorithm. The average of these maximum IoUs across all examples is the mIoU. If the maximum IoU is greater than a threshold, we consider it a true positive (TP), otherwise a false positive (FP). Thus, Recall is calculated as TP / (TP + FN) and Precision is calculated as TP / (TP + FP). These are averaged across all examples and across three thresholds (0.3, 0.5, 0.7) to get the mAR and mAP, respectively.
>
> Consider this example:
>
> Question: What processes unfolded from 213 to 512?"
>
> Prediction: "Hydrodissection from 213 to 272, Phacoemulsificiation from 272 to 512",
>
> GT: "Rhexis from 213 to 245, Hydrodissection from 245 to 293, Phacoemulsificiation from 293 to 512"
> 0.4166666666666667,1.0,0.5,0.5,0.6666666666666666,0.5,0.5
>
> There are 3 steps in the GT and prediction combined, namely Rhexis (s1), Hydrodissection (s2) and Phacoemulsification (s3).
> The IoU between the GT and predicted time range for each of them are:
>
> s1: 0 (since Rhexis is not in prediction)
>
> s2: (272-245)/(293-213) = 0.33
>
> s3: (512-293)/(512-272) = 0.91
>
> Hence, mIoU for this example would be 0.42.
>
> For a IoU threshold of 0.3, there are 2 TP, 1 FN and 0 FP. Hence, P@0.3 is 1.0 and R@0.3 is 0.67.
>
> Averaging this across 0.5 and 0.7 similarly, we get the complete set of metrics for this example. This is further averaged across all data examples to get the values in Table 1.
>
> Thus, in summary, the mAP measures the ability of the VLM to predict the correct boundaries, while the mAR measures the ability of the VLM to detect all relevant steps. The mIoU denotes how temporally accurate the model is for the detected steps.

---

> ### Author Response · Authors · 2026-01-22
> **2) Confusion regarding False Negative**
>
> We sincerely apologize for the typo in Section 4.3, where the correct line should be: "The ground truths that were not matched with any prediction are considered as false negatives, while the predictions that were not true positives were considered as false positives". We have updated it in the revised paper in the supporting documents. We can assure the reviewer that this was purely a writing error and that the implementation and all reported results use the correct definitions. In the example above, we show the metrics for an example, where we consider the step missed by the prediction as a false negative. The IoU is calculated according to the Hungarian algorithm that matches each prediction with the maximal IoU prediction. We observed that our model does not produce multiple predictions for the same step and time range since we do not utilize a Q-former based decoding approach that suffers from this issue.

---

> ### Author Response · Authors · 2026-01-22
> **3) Key annotation Detail for Cat-1K**
>
> Working with an expert surgeon, we prespecified definitions for the start and stop of each step. A common set of 50 videos was annotated by all annotators to verify agreement among them with a tolerance of 1 second for step boundaries. The remaining videos were annotated by at least one annotator, and a random 10% sample was annotated by two different annotators to verify inter-annotator agreement (for all videos).

---

> ### Author Response · Authors · 2026-01-22
> **4) Ablation Comparability**
>
> In Table 2, Ablation 3 shows our pipeline, with a single LoRA that handles all three tasks. This setup has VMAE features and the IM-TS Adapter and a single LoRA. We see that the performance is quite low, since without the disentanglement between tasks, the model is not able to generate the correct format for the output. In Table 3, Ablation 1, we use multiple LoRAs but without the VMAE features or IM-TS adapter and we see a similar performance. In Table 3, Ablation 2 and Ablation 3, we see usage of multiple LoRAs along with one of the components gives a clear benefit over the above two cases. Finally, the last row incorporates all the model components and design choices, giving the maximum benefit.

---

### Official Review · Reviewer_qPm4 · 2026-01-08

**Confidence:** 5
**Preliminary Rating:** 3
**Final Rating:** 3

**Summary:**

This paper presents CatVLM, a Vision Language Model for cataract surgery videos that improves temporal reasoning in surgical video understanding. It uses temporal boundary awareness by linking video clips to their timestamps, allowing the model to focus on specific moments instead of treating videos as unordered sequences. CatVLM is tested on three tasks: Video Moment Retrieval, Video Captioning, and Counting, using a new set of task-specific question–answer annotations. The work shows that modeling temporal boundaries helps VLMs better understand surgical videos and provides resources for future research in this area.

**Strengths:**

The paper presents a meaningful improvement over existing Vision Language Models (VLMs) for cataract surgery by explicitly incorporating temporal reasoning and adapting the model to different tasks using LoRA modules for task-specific fine-tuning. The integration of the im-TS adapter is another strong point, as it further enhances the model’s ability to capture temporal dynamics in untrimmed videos. This design allows CatVLM to handle multiple clinically relevant tasks, such as video moment retrieval, captioning, and counting, within a single framework, increasing its practical utility. The attempt to create a fully automatic system for moment-level reasoning is promising, even though there is room for improvement. The manuscript is clearly written, well-structured, and situates the work appropriately within existing literature, making it accessible and scientifically sound. Figure 2 effectively visualizes the complex model architecture, helping readers understand the workflow and interactions that would be difficult to convey through text alone. Overall, the paper demonstrates scientific merit and potential value to the surgical AI community by combining methodological innovation, task adaptability, and clear presentation.

**Weaknesses:**

The paper introduces a task-adaptive approach using multiple LoRA modules and a KNN-based system for selecting the appropriate LoRA, but these challenges are common in task-incremental and continual learning, such as catastrophic forgetting, and the paper does not reference or evaluate alternative strategies from that literature. The KNN method achieves high reported accuracy, but no visualization of the latent space or embeddings is provided, leaving the effectiveness and robustness of task separation unclear. The requirement to manually generate and embed 50 synonymous sentences per question may limit scalability and generalizability, and alternative approaches could have been explored to address this more efficiently. The evaluation focuses on relatively simple or constrained questions, which limits the demonstration of CatVLM’s advantages over classical CNN-RNN architectures, as they are very close in the respective task they are solving, even a direct comparison is impossible. In fact, performance on the Cataract-1K dataset (See table 6 of the paper) of standard CNN-RNN models have achieved appears much higher even tho the clinical value of the models on the evaluated questions is almost the same, raising questions about the practical benefit of the proposed method. Making a bigger distinction of the unique benefits of VLM approach over classical DL model would have been great and may would have been achieved by incorporating more advanced questions that can be asked.

**Detailed Comments:**

The author used CATVLM vs CatVLM in 3.1.

On the second page seem to be no citations even so quite some related works are first presented/reference in the introduction they do only get their citations later in the text. For reading flow, it would be great if they are cited on first occurrence.

**Justification Of Final Rating:**

The work tackles an important problem and presents a well-structured, thoughtfully designed VLM framework with appealing modularity and extensibility. The authors’ responses and added analyses are appreciated and clarify the motivation and scope of the approach. However, the current evaluation does not yet convincingly demonstrate clear practical or clinical advantages over simpler CNN/RNN-based alternatives, nor does it fully showcase the claimed benefits of open-set reasoning and task flexibility. Overall, the paper is promising, but the evidence remains insufficient to clearly support its central claims at this stage.

**Justification Of The Preliminary Rating:**

The work addresses an important problem in surgical video understanding and proposes a thoughtful task-adaptive VLM approach with LoRA modules and temporal reasoning, which is well-motivated and clearly presented. The modular design, integration of temporal boundaries, and generation of task-specific QA annotations are all positive contributions, and the writing and structure are strong. However, the paper does not clearly demonstrate the unique advantages of the VLM approach over classical CNN-RNN models, especially on the Cataract-1K dataset, where performance is comparable or lower. The evaluation focuses on relatively simple questions, leaving the potential benefits of moment-level reasoning and VLM capabilities underexplored. Overall, the methodological contribution is solid, but the paper falls short in convincingly “selling” why this approach is distinctly better than standard approaches.

**Questions To Address In The Rebuttal:**

Can the authors provide additional evidence that the KNN-based LoRA selection reliably separates tasks, for example through visualizations of the latent space or embeddings, and clarify how robust this method is to (adversial) variations in the input questions?

The approach requires generating and embedding 50 synonymous sentences per question for KNN training. Are there alternative, more scalable approaches that could be used, and how might these impact performance and generalizability to new tasks or question types?

Given that performance on the Cataract-1K dataset appears lower than standard CNN-RNN models and that the current questions are relatively simple, can the authors clarify the unique practical advantages of the VLM approach, and would incorporating more complex or clinically relevant questions better demonstrate these benefits?

---

> ### Author Response · Authors · 2026-01-22
> **1) KNN visualization**
>
> We thank the reviewer for their detailed comments and try to address them pointwise. We request the reviewer to kindly raise their score if we were able to address their concerns satisfactorily
>
> We have included a file named pca.png in the Supporting Documents where we use Principal Component Analysis (PCA) to illustrate how the KNN separates question types in the embedding space. In this figure, we see a clear separation between the VC questions that do not have step name (type 2) versus MR and Counting questions that have the step name (type 1 and 3 in figure). Within these two, there is a comparatively thinner difference, which causes the inaccuracy of 3%.
>
> It is important to note that KNN is a very simple method which can facilitate automatic LoRA selection. However, we believe that for most practical purposes, it is very easy to simply select the LoRA to be used manually, removing the need for any classifier mechanism. We do agree that KNN is a simple method and that better methods exist that can handle adversarial attacks better. However, kindly note that we introduce the KNN classifier as a proof of concept and it is not the main contribution of the paper. As noted by the reviewer, our main goal is to enhance temporal awareness of the VLM.

---

> ### Author Response · Authors · 2026-01-22
> **2) KNN scalability**
>
> To generate the training data for the KNN, we prompted ChatGPT to generate N sentences which are synonymous and non-repeating, given the sentences used to train CatVLM. The generated sentences could be used to train the KNN. Here, we used N=50, but this number can be scaled to larger numbers and for more tasks, given the current power of LLMs. Another approach could be to ask a large pretrained LLM to classify the question type among the existing tasks instead of training one's own classifier. This approach is training free and generalizable to most tasks without any data generation. Batch processing of questions can be done to increase scalability and decrease the amortized cost of using the pretrained LLM. We explored using KNN as a simple proof of concept to show that automation in LoRA selection can be achieved. However, we believe that manual selection of LoRA is a very feasible alternative.

---

> ### Author Response · Authors · 2026-01-22
> **3) Practical Advantages of VLMs over CNN/RNNs**
>
> There is not a lot of literature on temporally aware LLMs in the medical domain yet and we hope our work encourages further research that also beats CNN/RNN performance. However, we believe that in the current state also, there are practical advantages of using VLMs because of the following:
>
> (i) Handling open-set questions: Users can directly ask CatVLM questions in the natural language. CNN/RNN-based methods would require modifying the input to a fixed structure and vocabulary.
>
> (ii) Easy scalability to more tasks: In CatVLM, adding new tasks is as simple as adding a new LoRA module. This makes it possible to scale it using continual learning and collaborative efforts where researchers can add their LoRA modules for different tasks to a common repository. On the other hand, CNN/RNN approaches typically require designing and training an entirely new architecture for each task.
>
> We agree that more complex clinical questions are a promising next step, and CatVLM’s architecture (task‑specific LoRAs + timestamp‑aware features) is explicitly designed to support such extensions.

---

> > ### Comment · Reviewer_qPm4 · 2026-01-25
> > **Thanks for the answers**
> >
> > It is true that VLMs and LLM-based systems offer greater flexibility than rigid CNN/RNN architectures when it comes to open-ended queries and task extensibility. However, in the current evaluation, this advantage is not convincingly demonstrated. None of the evaluated questions or tasks require a level of openness or reasoning that goes meaningfully beyond what could be achieved by feeding structured CNN/RNN outputs into a downstream language model, so the claimed open-set capability does not translate into a clear information gain in practice.
> >
> > Similarly, while the modular LoRA-based design is attractive from an engineering and scalability perspective, the experiments do not show that this flexibility leads to improved clinical usefulness or qualitatively different behavior compared to simpler alternatives. Without tasks that genuinely require cross-task reasoning, abstraction, or complex temporal understanding, it remains unclear whether the proposed VLM architecture offers advantages beyond ease of integration. Overall, the work is promisingly great, but the evaluation remains weak in demonstrating clinical value and does not yet convincingly support the paper’s central claims. The newly added PCA figure is appreciated and helps clarify the learned feature structure.

---

> > > ### Author Response · Authors · 2026-01-26
> > >
> > > We agree that adding more complex tasks that require richer temporal reasoning and more complex QA structures would help show the clinical significance more. While this is indeed a limitation of the current evaluation scheme, we believe that it is more of a data constraint than an architecture constraint. As mentioned in the paper, the current datasets are created using derivatives of pre-existing datasets that had expert annotations for step boundaries. Currently, to the best of our knowledge, there is no dataset in the cataract domain that has such QA pairs that requires rich temporal understanding.
> > >
> > > One way to create such a dataset would be to use a large-scale VLM. However, without proper adaptation to the cataract domain, it would definitely lead to a lot of hallucinations. The other, more tedious and straightforward approach would be to manually annotate, which is expensive. We tried creating an approach that would keep evolving with more such data and tasks being created, but at the current evaluation stage, we agree with the reviewer that more complex tasks can improve clinical applicability.
> > >
> > > We thank the reviewer for also acknowledging the strengths of the paper and for the constructive feedback that will help guide the next stage of this research. We fully respect the reviewer’s final assessment and sincerely appreciate the time and care invested in evaluating our work.

---

### Official Review · Reviewer_FoYa · 2026-01-15

**Confidence:** 5
**Preliminary Rating:** 5

**Summary:**

This study proposed CatVLM, a temporal boundary-aware Vision Language Model that learns to represent fine grained temporal dynamics in untrimmed cataract surgery videos.
CatVLM facilitates moment level reasoning of 3 clinical tasks, namely Video Moment Retrieval, Video Captioning, and Counting.
Based on task-related QA annotations and timestamp-integrated video clips, this study evaluated CatVLM on two publicly available datasets, setting good baselines and demonstrating that explicit modeling of temporal boundaries is crucial to medical video perception.

**Strengths:**

The article introduces a new boundary sensitive VLM of cataract surgery videos, which fills a significant gap in the temporal reasoning. It is properly organized, well-written, scientifically grounded, and can be useful in the further development of medical video knowledge and research

**Weaknesses:**

This paper indicates that there are limited performance improvements on a few tasks with the obvious overfitting on video captioning and moment retrieval.

It is also only evaluated using cataract datasets, which limits the ability to apply the results to other areas of surgery.

**Detailed Comments:**

Nil

**Justification Of The Preliminary Rating:**

This paper presents a new method of temporal reasoning in surgical videos that has a sound methodology and assessment.

Although the performance, though not always high, is favourable in all tasks, the input is significant and has potential in future studies.

**Questions To Address In The Rebuttal:**

Nil

---

> ### Author Response · Authors · 2026-01-22
> **Response to Reviewer FoYa**
>
> We thank the reviewer for their appreciation of the paper. We will extend the work in the future for more tasks and domains

---

### Author Rebuttal · Authors · 2026-01-22

**Rebuttal:**

We thank all the reviewers for their valuable comments. We are encouraged that the reviewers found our approach to enhancing the temporal awareness of VLMs for cataract surgery analysis to be a meaningful and valuable contribution. The primary concerns raised were regarding the task formulation and the automatic LoRA selection mechanism. These have been addressed in detail in the “Official Comments” section individually.  Additionally, all minor changes suggested by the reviewers have been added to the revised paper and highlighted. The revised paper can be found in the Supporting Material.

We request the reviewers to kindly raise their scores if we were able to address the concerns in a satisfactory manner,

**Supporting Material:**

/attachment/17ac8b91c93d1bda84357e8f41797d2670911a4a.zip

---

### Meta-Review · Area_Chair_4iqh · 2026-02-01

**Recommendation:** Accept (Poster)
**Confidence:** 4

**Metareview:**

This paper proposes CatVLM, a boundary-aware vision language model for temporal reasoning in cataract surgery videos, targeting clinically relevant tasks such as video moment retrieval, interval-based narration, and counting. Reviewers agree that the problem setting is important and timely, and that explicitly modeling temporal boundaries via timestamp-aware representations and task-specific LoRA adapters is a reasonable and well-motivated design choice. The paper is clearly written, technically sound, and supported by extensive experiments, ablations, and qualitative results on two public datasets. The authors have responded carefully to reviewer concerns by clarifying task formulations, correcting evaluation definitions, and improving transparency around metrics, annotation protocols, and ablation comparability. At the same time, several reviewers note that the current evaluation focuses on relatively constrained question types and does not yet convincingly demonstrate clear practical or clinical advantages over simpler CNN or RNN-based alternatives, nor does it fully exploit the open-ended reasoning capabilities of VLMs. These limitations temper the strength of the contribution and make the work less suitable for an oral presentation. Overall, the paper presents a solid and promising step toward temporally aware medical video understanding and will be of interest to the MIDL community, and is therefore recommended for acceptance as a poster.

---

### Decision · Program_Chairs · 2026-02-13

Accept (Poster)